# Universal Stress Proteins: From Gene to Function

**DOI:** 10.3390/ijms24054725

**Published:** 2023-03-01

**Authors:** Dan Luo, Zilin Wu, Qian Bai, Yong Zhang, Min Huang, Yajiao Huang, Xiangyang Li

**Affiliations:** National Key Laboratory of Green Pesticide, Key Laboratory of Green Pesticide and Agricultural Bioengineering, Ministry of Education, Guizhou University, Guiyang 550025, China

**Keywords:** multi-functional roles, molecular mechanism of USPs, universal stress protein (USPs)

## Abstract

Universal stress proteins (USPs) exist across a wide range of species and are vital for survival under stressful conditions. Due to the increasingly harsh global environmental conditions, it is increasingly important to study the role of USPs in achieving stress tolerance. This review discusses the role of USPs in organisms from three aspects: (1) organisms generally have multiple USP genes that play specific roles at different developmental periods of the organism, and, due to their ubiquity, USPs can be used as an important indicator to study species evolution; (2) a comparison of the structures of USPs reveals that they generally bind ATP or its analogs at similar sequence positions, which may underlie the regulatory role of USPs; and (3) the functions of USPs in species are diverse, and are generally directly related to the stress tolerance. In microorganisms, USPs are associated with cell membrane formation, whereas in plants they may act as protein chaperones or RNA chaperones to help plants withstand stress at the molecular level and may also interact with other proteins to regulate normal plant activities. This review will provide directions for future research, focusing on USPs to provide clues for the development of stress-tolerant crop varieties and for the generation of novel green pesticide formulations in agriculture, and to better understand the evolution of drug resistance in pathogenic microorganisms in medicine.

## 1. Introduction

In 1992, Nyström and Neidhardt [1] identified a protein in *Escherichia coli* that was overexpressed under stressful conditions and named it Universal Stress Protein (USP) [2,3]. The USP was shown to play an important role in various [4] aspects of phosphorylation [5] and coordination of glucose [6] and acetate metabolism in *E. coli* [7]. After the amino acid sequence of the first USP was determined in *E. coli* [8], USPs were identified in a range of organisms [9,10]. The *E. coli* USP gene family includes *UspA, UspC (YecG), UspD (YiiT), UspE (YdaA), UspF (YnaF, UP03) and UspG* (*YbdQ)*, and they have been divided into three subfamilies (Figure 1a) [11]. Various *E. coli* USPs play different roles in this bacterium [12] and can be divided into two major classes (Figure 1b) and four minor subclasses (Figure 1c) [13].

Follow-up studies have found that USP genes are present not only in *E. coli* [14], but also in the genomes of other bacteria [15], archaea, fungi, protozoa [10], and plants [16]. It was found that the USP genes are involved in the formation of biofilms [17] that help bacteria survive in an anaerobic environment [18]. *Oryza* sativa was the first eukaryote in which a USP protein was reported, and OsUSP1 was shown to be particularly closely related to the bacterial MJ0577-type ATP-binding USP protein, possibly playing a role in ethylene-mediated anaerobic stress adaptation in rice [19]. Subsequent studies [20] have detected USPs in a number of plant [21] species with expression upregulated in response to stresses [22] such as drought [23,24] or cold [25], as well as with plant root nodule formation [26]. Studies at the molecular level [27] have shown that USPs are associated with a range of functions, such as protection of nucleic acids, cellular defense [28], stress tolerance [29], protein scaffolding, and cellular protein transport [30]. The study of USPs in plants is not as in-depth as in microorganisms, and most of the studies on plant USPs focus on plant resistance studies, with few studies on structure and other aspects. In this paper, we summarize the studies on USPs from three aspects: genetic, structural and functional, in order to provide a reference for the study of USPs in plants.

## 2. Diversity of USP Genes

The *Acinetobacter baumannii* universal stress protein A (USPA) protects the bacterium, a major global public health threat, from H_2_O_2_ stress [31]. *Schistosoma mansoni* is the causative agent of human schistosomiasis, and all *S. mansoni* USP genes are transcribed during at least one developmental stage of the helminth life cycle, with the expression of six of these genes being upregulated in the trichocysts, a free-swimming developmental stage of *S. mansoni* that is essential for transmission to the intermediate snail host. After transmission into the intermediate host, the *S. mansoni* USP transcripts may be induced to perform specific functions triggered by environmental stressors [32].

*USP* genes can be an important marker for studying species evolution. Espinola et al. provided a broad framework for the evolution of the *USP* gene family to provide a basis for future studies on the emergence of USPs in other tax [33]. A recent study of the gene encoding the allosteric universal stress protein (USPA) in *Halomonas* spp. has demonstrated its role in microbial evolution, its presence suggesting that *Halomonas* spp. developed directly from primitive bacteria [34]. Therefore, the study of microbial USPs can help us understand the origins and mode of action of these proteins, which would be beneficial to the human exploitation of microbial resources.

The model plant Arabidopsis thaliana possesses as many as 41 *USP* genes (Table 1), which are extremely important in the developmental process of this plant [35] *USP* genes play specific roles at each developmental stage; different genes can also respond when the plants are subjected to different stresses [36]. For example, barley has *HvUsp* genes that are specifically expressed only in roots and leaves [37]. The AtUsp genes positively regulates genes encoding cell wall components, thus facilitating cell expansion during organ growth [38]. The *AtUsp* promoter can be highly induced by plant hormones and a number of abiotic stresses and can also be induced by other effectors to reduce the effect of stress on important traits in the plant [39]. The expression of At3g58450 is regulated by phytohormones; it is involved in *A. thaliana* seed germination and may also regulate the flowering process [36,40]. The expression of *AtUsp17* can be induced by a number of stresses and this gene regulates salt tolerance in A. thaliana [41].

Studies targeting the 21 USPs in grape (*Vitis vinifera*) revealed that the *VvUspA* promoter contains potential hormone response and stress-related elements, suggesting that the *VvUspA* gene may be involved in various hormone and stress response pathways [42]. A full-length 678-bp cDNA fragment, containing a *MsUsp* 528-bp coding sequence, was identified within Medicago sativa (alfalfa) leaves, and this gene was found to be commonly expressed in leaves, stems, flowers, roots, rhizomes, and seeds, with the highest expression levels being in seeds and relatively low levels in flowers and roots [43]. The *Solanum pennellii* gene (*SpUsp*) is located on chromosome 1 with a length of 572 bp, including a 438-bp open reading frame (ORF), a 91-bp 5′-untranslated region, and a 43-bp 3′-untranslated region, and has been found to be a drought-responsive gene [44]. A total of 44 USP genes, ranging in length from 222 bp (*OsUsp31*) to 2817 bp (*OsUsp20*), have been identified in rice, located on 11 of the 12 chromosomes, with the exception of chromosome 4, and alternative splicing of the primary transcript has been recognized, enabling the synthesis of multiple proteins from a single gene [45]. *Malus sieversii* contains a *MsUspA* gene of length 222 bp, and its expression decreases reactive oxygen species (ROS) accumulation and enhances plant drought tolerance [46]. The analysis of 16 *HvUsp* genes in barley revealed that each gene contained two or three introns, which range in size from 75 to 941 bp, and expression is upregulated in response to salt stress [37]. There are 71 genes encoding USP-related structural domains in pigeonpea (*Cajanus cajan*) [47], 49 of which have been shown to be drought-responsive genes [48]. The *GaUsp1* [49] gene in Gossypium arboreum is associated with drought response [50], whereas the *GaUsp2* gene is associated with salt tolerance and osmotolerance [51]. The full-length sequence of *SbUsp* cDNA from the extreme halophyte Salicornia brachiata is 873 bp long, containing a 486-bp ORF, and the *SbUsp* gene promotes plant growth, reduces ROS accumulation, maintains ion homeostasis, and improves plant physiology under conditions of salt or osmotic stress [52]. This species was also found to contain two homologs of the intronless *SbUsp* gene, which encode salt- and osmosis-responsive USPs [53]. The medicinal plant Calotropis procera is a good model plant for the study of salt- and drought-tolerance genes, and the discovery of the USP gene of this plant is of great significance for the selection and breeding of anti-adverse varieties [54]. *Salvia miltiorrhiza*, a highly prized plant in traditional Chinese medicine, contains 32 cDNAs of USP family members, ranging in length from 279 bp (SmUSP11) to 2274 bp (SmUSP3) [55]. *MaUsp* genes identified from mulberry (Morus alba L.) are capable of conferring cellular-level tolerance in both prokaryotic and eukaryotic systems. This finding led to a new direction of thinking for subsequent USP studies, namely that maintaining cell-level tolerance is essential to maintaining growth under stress and that cell-level tolerance can be enhanced by overexpression of genes such as USPs [56].

## 3. USP Crystal Structure

Thomas et al. (1998) isolated and purified MJ0577 from the hyperthermophilic bacterium *Methanococcus jannaschii,* and resolved its protein crystal structure (1MJH, PDB DOI: 10.2210/pdb1MJH/pdb). The MJ0577 monomer consists of five-stranded twisted parallel β-sheet lamellae, with two on each side of the lamella helix, which can bind ATP, although the protein itself is not an ATPase, but exercises the function of an ATPase in physiological activities or as an ATP-mediated molecular switch in vivo [57]. Marcelo et al. (2001) resolved the crystal structure of the USPA of *Haemophilus influenzae* (1JMV, PDB DOI: 10.2210/pdb1JMV/pdb), which has a tertiary structure similar to that of the α/β-fold of MJ0577, but which does not bind ATP. The protein is present in solution as a dimer, with sulfate ions forming the dimer by chelating with the side chains through Arg28 and His29 [58]. Mycobacterium spp. stress protein MSMEG-3811 (5AHW, PDB DOI: 10.2210/pdb5AHW/pdb) binds cyclic adenosine monophosphate (cAMP), with the monomeric structure of this protein having a typical, open, twisted five-stranded parallel β-sheet layer with a topology of β3-β2-β1-β4-β5 sandwiched between six helices of different lengths, with a binding pocket that acts as a protein regulator of downstream effectors of cAMP-binding proteins, so that the abundance of this protein may also determine the amount of free cAMP in the cell [59]. In the same year, the crystal structures of YdaA (USPE, 4R2J, PDB DOI: 10.2210/pdb4R2J/pdb) and YnaF (USPF, 4R2L, 10.2210/pdb4R2L/pdb) of Salmonella typhimurium were resolved for the first time [60], and the monomer of YnaF was shown to consist of five chains (S1–S5) with the following chain order S3-S2-S1-S4-S5, the chain ends being connected by α-helices. The two YnaF plasmids form an asymmetric unit that is stabilized by interaction with chloride ions to form a tetramer with symmetry. YdaA is a larger protein than YnaF and the YdaA protomer includes an N-terminal structural domain (residues 1–149) and a C-terminal structural domain (residues 150–315) in addition to two tandem USP structural domains. The two protomers of YdaA combine to form a tetramer similar to the tetrameric structure of YnaF. To verify the functions of YdaA and YnaF, each of these two proteins was mutated and the crystal structures of the mutant proteins 4R2K (PDB DOI: 10.2210/pdb4R2K/pdb) and 4R2M (PDB DOI: 10.2210/pdb4R2M/pdb) were obtained [60].

NE1028 (3TNJ, PDB DOI: 10.2210/pdb3TNJ/pdb) from Nitrosomonas europaea has potential ATP-binding residues, and the structure of the complex formed by its binding to AMP was reported by Tkaczuk et al., who also reported that the mutant 2PFS (PDB DOI: 10.2210/pdb2PFS/pdb), namely the universal stress protein 3QTB complexed with dAMP from Archaeoglobus fulgidus(PDB DOI: 10.2210/pdb3QTB/pdb), and the universal stress protein 6HCD (PDB DOI: 10.2210/pdb6HCD/pdb) from archaea [30]. The histidine kinase KdpD in the KdpDE two-component system (TCS) contains a USP structural domain that binds to the second messenger cyclic diadenosine monophosphate (c-di-AMP) and is used to regulate the transcriptional output of the TCS in thick-walled bacterial taxa such as Staphylococcus aureus. Given this structure, Dutta et al., suggested that the USP structural domain in the KdpD histidine kinase may represent a new USP subfamily [61]. The crystal structure of *E. coli* USPE (5CB0, PDB DOI: 10.2210/pdb5CB0/pdb) was resolved [62], showing that USPE folds into a fan-shaped structure and has a hydrophobic cavity bound to its ligand [63]. The Arabidopsis protein At3g01520 (2GM3, PDB DOI: 10.2210/pdb2GM3/pdb) is the only eukaryotic universal stress protein crystal structure that has been resolved, and the structure shows that it is an aggregate and that each monomer is bound to an AMP molecule its [64].

The crystalline small molecules containing ATP or ATP analogues (AMP, ANP, etc.) were selected for sequence comparison and structure superposition, and the results are shown in Figure 2.

It is well known that structure determines function, and by combing through them we found that most USPs structures rarely exist as single chains when they are resolved, (Table 2) they are often aggregated in even chains, while ATP is involved in protein crystallization, which may be an important reason for its function in plants.

## 4. Functional Diversity of USPs

### 4.1. Functional Diversity of Prokaryotic USPs

USPs are important regulatory stress proteins which have been reported from a wide range of bacterial species [65], which help bacteria survive under conditions of stress. USPs have various functions in fine bacteria, such as ATP hydrolysis [66], modification of membrane properties [67] and chlorine sensing [68]. *Mycobacterium tuberculosis* expresses USPs to survive hypoxia and carbon monoxide stress, and it was found that this bacterium expresses ten USPs, which were divided into five classes (Figure 3a) [69]. Universal stress proteins from *M. tuberculosis*, such as Rv2623 and Rv2624c, bind ATP [27], whereas Rv1636 binds more to cAMP than to ATP [70]. Rv2623 regulates mycobacterial growth both in vivo and in vitro (interacting with Rv1747; Figure 3b), suggesting that it is an essential protein for M. tuberculosis during the chronic phase of host infection. It was found that the ATP-binding activity of Rv2623 determines the growth-regulating properties of the USP and that Rv2623 may act as an ATP-dependent signaling intermediate during persistent infection of the host [71]. Rv2624c alters the abundance of arginine in vivo by binding to ATP, to make M. tuberculosis more likely to survive in the host [72]. At the same time, overexpression of the universal stress protein BCG-2013 in M. tuberculosis, which is associated with latency, increases the bifunctional catalase-peroxidase KatG level, which makes the overexpressed strain more sensitive to isoniazid (INH) [73]. Species in the bacterial family *Desulfovibrionaceae* have 651 sequences associated with the expression of USPs, which can be divided into four classes based on the number of structural domains and ATP-binding motifs (Figure 3c) The USPs are associated with the survival of members of the *Desulfovibrionaceae* in anaerobic aquatic environments and also with cellular uptake of inorganic mercury and the production of methylmercury [74].

USPs may be a major regulator of bacterial survival, and it has been suggested that the USP of Micrococcus luteus may be a switch for metabolic control in this bacterium, with related studies suggesting that this USPA616 regulates glyoxylate shunting to make it more likely for the bacterium to survive under stressful conditions [65]. USP4207 from Mycobacterium smegmatis is closely involved in biofilm formation, with strains lacking USP4207 exhibiting reduced biofilm formation in vivo, resulting in coarser colony morphology [43]. The presence of USP76 in lung cells is a hallmark of *Burkholderia* infection [75]. *Listeria innocua* ATCC 33090 contains a novel ATP-binding USP that is upregulated not only during the stable phase but also during the exponential growth period, and it has been found to be involved in the in vivo bacterial response system when acid stress is encountered during the exponential period [76]. USPF is involved in the tolerance process of atypical enteropathogenic *E. coli* (aEPEC) and other Enterobacteriaceae [77].

USPs are associated not only with bacterial stress tolerance but also with the pathogenicity of pathogenic bacteria, and are important factors for the survival or persistence of various pathogens [78,79]. *A. baumannii* USPA plays an important role in the pathogenicity of the two most lethal, infectious diseases caused by this bacterial human pathogen, pneumonia and sepsis, and it is an essential component of the virulence mechanism of *A. baumannii* [31]. *Edwardsiella piscicida* USP13 is essential for pathogenicity and can help block the host immune response to pathogen infection [80]. USPC was found to function as a scaffold for signaling in the KdpD/KdpE-P/DNA complex [81] and to regulate the expression of the ion pump/channel complex kdpFABC [82], the N-terminal domain of YdaA binds zinc and might play a role in lipid metabolism [60].

### 4.2. Functional Diversity of Eukaryotic USPs

USPs also carry out multiple functions in eukaryotic organisms, and *Schistosoma mansoni* USPs may play a role [83] in defense against hydrogen peroxide-induced oxidative stress [32], so that USPs are now novel targets for human schistosomiasis intervention and treatment [84]. USPs, as important stress-regulated proteins in plants, are involved in a variety of physiological activities.

#### 4.2.1. Versatility of *A. thaliana* USPs

There are 53 USPs in Arabidopsis, and these USPs are divided into four groups (Table 1). They are expressed in almost all parts of the plant and show tissue specificity and different functions at various developmental stages [35]. AT3G53990 has several possible functions in Arabidopsis. AT3G53990 transforms itself from a low molecular weight (LMW) complex to a high molecular weight (HMW) complex in response to high temperature scorching stress, and such a structural change could help plants protect key intracellular proteins in high temperature environments, and it would participate as a protein chaperone to play a stabilizing role as a protein [85] (Figure 4a). AT5G35380 is an effector of proline accumulation at low water potential [86]. The GRUSP protein, encoded by the AT3G58450 gene, is a novel regulatory component of the flowering signal transduction pathway in Arabidopsis, and overexpression of this protein interferes with the flowering signal, so that flowering is delayed, and a decrease in the concentration of the endogenous, bioactive gibberellins GA1 and GA3 is detected [40]. GRUSP is also associated with Arabidopsis seed germination and has a similar regulatory pattern to the hormone, but unlike abscisic acid, it promotes seed germination [36]. HRU1 (Hypoxia-responsive universal stress protein) coordinates oxygen sensing and ROS signaling under hypoxic conditions [87]. AtUSP17 negatively mediates salt tolerance in Arabidopsis by regulating ethylene, ABA, ROS, and G-protein signaling and responses [41].

As important stress-regulated proteins, USPs are involved in a range of physiological activities in plants. It has been shown that cold-shock tolerance is usually achieved by RNA chaperones [88], and that, in Arabidopsis, AtUSP becomes involved in plant physiological activities as an RNA chaperone, which helps RNA-bound proteins to exercise their functions correctly by preventing RNA misfolding or by resolving misfolded RNAs [89] (Figure 4b). Meanwhile, when subjected to environmental stress, the expression of USPs causes upregulation of the expression of secondary metabolites in plants, resulting in an increase in intracytoplasmic solutes and achieving a greater tolerance to stress [46]. More importantly, USPs can also reduce ROS production in plants [90], maintain ROS homeostasis, alleviate oxidative damage caused by ROS, and improve tolerance to oxidative stress [91]. Interestingly, Arabidopsis AtUSP (At3g53990) exhibits anti-fungal activity by generating ROS and causing mitochondrial damage in the pathogenic fungi [92]. This suggests that the study of plant USPs could be beneficial for the selection of disease-resistant varieties and the creation of novel pesticides.

Plants have evolved complex redox signaling [91] regulatory systems [93] and can tolerate stress by activating specific intracellular redox-mediated defense signaling pathways [94,95]. The plant redox system was found to regulate ROS concentration in plants in relation to cell metabolism [96,97], apoptosis [98,99], and carbon metabolism and photosynthesis-related processes [100,101]. It has been shown that three Arabidopsis USP proteins, HRU1, *At*USP, and At3g17020, interact with their redox chaperone thioredoxin-h1, and that the structural transition of *At*USPs is induced by external redox changes, accompanied by changes in their function.

#### 4.2.2. Functional Diversity of USPs in Other Plants

The tomato USPA (SlRd2) is an ATP-binding protein that forms homodimers in plants. SlRd2 is a novel interactor and phosphorylation target of SlCipk6, a member of the CIPK (CBL-interacting protein kinase) family, and functionally regulates SlCipk6-mediated ROS generation [102]. Under drought conditions, overexpression of the *S. pennellii* gene (SpUsp) causes upregulation of a large number of chlorophyll a- and b-binding proteins and photosystem proteins in plants, increasing the ABA concentration, closing stomata, alleviating oxidative damage caused by ROS, and improving tolerance to oxidative stress [44].

The expression and regulation of *Gossypium arboretum* GUSP1 under drought stress results in drought tolerance in this plant [50]. It has been demonstrated that, following the induction of drought tolerance, the relative water content, total chlorophyll content, CO_2_ absorbed by net photosynthesis, stomatal conductance, total soluble sugars, and proline concentration of the plant’s leaves increased significantly whereas the relative membrane permeability and transpiration rate decreased significantly, suggesting that GUSP1 may activate some downstream genes in signal transduction pathways in response to drought or other abiotic stresses in order to protect membranes and cells from damage [49]. The results of the mutant assay of heterologous expression of GUSP-2 with a lysine-to-threonine mutation demonstrate that this protein may be directly involved in stress tolerance or may act as a signaling molecule to activate stress adaptation mechanisms [51]. A universal stress protein in *M. sieversii* (MsUSPA) is involved in the regulation of hormone and secondary metabolite synthesis to reduce transpiration and retain water by altering the cellular structure of leaves to improve drought resistance. In addition, overexpression of MsUspA increases the activity of antioxidant enzymes and improves antioxidant capacity, reducing the accumulation of ROS [46].

Of the 32 members of the *S. miltiorrhiza* SmUSP protein family, four target mitochondria, four target chloroplasts, and two specifically compete for the secretory pathway [55]. The *S. brachiata* SbUSP is a membrane-bound cytoplasmic protein that interacts with AMP and exhibits characteristic phosphorylation and glycosylation motifs and ATP-binding sites, suggesting that SbUSP may be directly involved in tolerance mechanisms or act as a molecular switch (signaling molecule) to activate stress adaptation mechanisms [53]. The archaea *Sulfolobus acidocaldarius* contains *Sa*USPA, which is an ATP-binding protein that binds to the phosphatase PP2A in vitro and in vivo. Although *Sa*USPA does not hydrolyze ATP, it stimulates the phosphatase activity of PP2A and may affect many other processes in this way [103]. The leguminous plant *Astragalus fridae* increased the expression rate of USP in response to exposure to *SiO_2_* nanoparticles [104].

## 5. Future Perspectives

It is clear that the study of USPs has moved from superficial investigations to in-depth examination of the mechanisms involved. In the future, the study of USPs may focus on the following trends: (1) despite their importance, our understanding of the structure of USPs is still insufficiently detailed, particularly that of eukaryotic USPs; (2) although USPs are able to help cells survive under stressful conditions, these proteins follow two completely different mechanisms, namely ATP-dependent USPs play a role in cellular transport, whereas ATP-non-binding USPs may function in the cell cycle [105]; (3) the application of USP research to agriculture may facilitate the development of stress-tolerant crop varieties, and the creation of new green pesticide formulations, using USPs as targets or lead compounds for therapeutic or protective agents; and (4) in the medical field, the study of USPs in drug-resistance mechanisms in pathogenic bacteria is important for the screening of novel therapeutic drugs [79] and the development of related antibodies and vaccines for the prevention and treatment of human diseases. USPs from different plants, which play different roles in vivo (Table 3), are also worthy of research in the future by genetically engineering plants to adapt them to more complex climatic challenges.

## Figures and Tables

**Figure 1 ijms-24-04725-f001:**
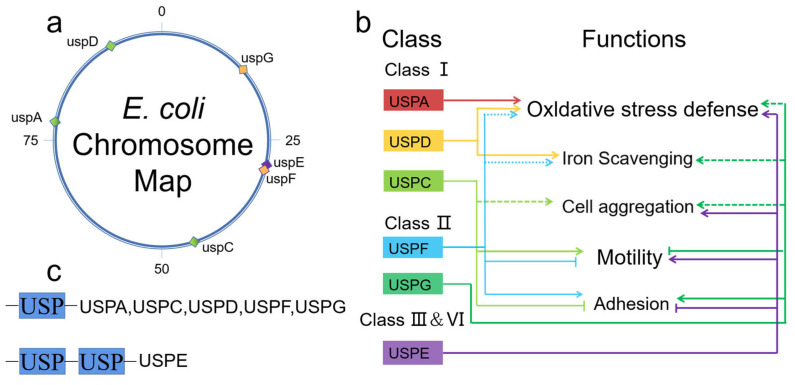
*E. coli* genes and functions. (**a**) Location of USP family genes on *E. coli* chromosomes. uspA, uspC and uspD belong to a category represented by green squares, uspF and uspG belong to another category represented by yellow squares, and uspE is represented by purple; (**b**) the role of six *E. coli* USPs in oxidative stress defense, iron metabolism, and cell surface properties; (**c**) classification of universal stress proteins of *E. coli*.

**Figure 2 ijms-24-04725-f002:**
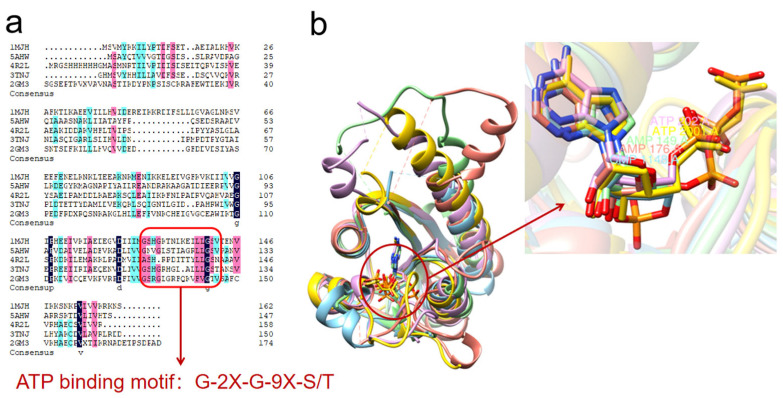
USPs binding analysis. Crystalline small molecules containing ATP or ATP analogs (AMP, ANP, etc.) were selected for sequence comparison and structure superimposition. (**a**) Sequence comparison. All except 4R2L contain ATP-binding motifs: G-2X-G-9X-S/T. (**b**) Superposition. 1MJH, 5AHW, 4R2L, 3TNJ and 2GM3 are gold, blue, purple, green and red, respectively. Superposition of 1MJH as the reference structure revealed that 4R2L still binds ATP at the same position even without the typical ATP binding pattern.

**Figure 3 ijms-24-04725-f003:**
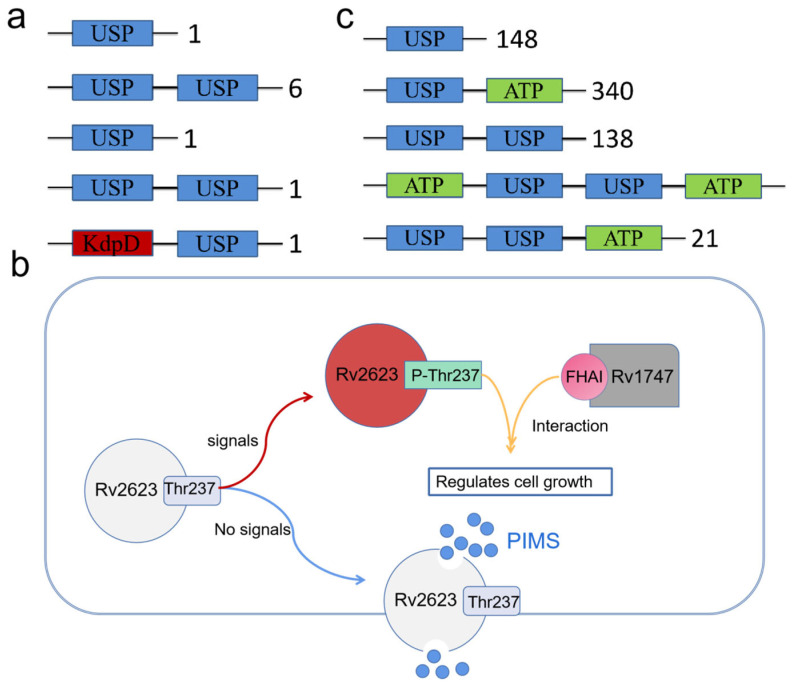
Classification of *M. tuberculosis*, Vibrio desulfuricans and interaction of Rv2623 with Rv1747. (**a**) There are ten USPs of *Mycobacterium nucleatum,* which are classified into five categories. (**b**) Rv2623 regulates *M. tuberculosis* growth. When certain signals are present, Rv2623f Thr237 is phosphorylated and interacts with the FHA structure of Rv1747; when there is no signal from Mycobacterium tuberculosis, this protein transports phosphatidylinositol mannoside (PIM), making *M. tuberculosis,* possess more virulence. (**c**) Classification of the USP of Vibrio desulfuricans. The USPs can be classified into five categories according to the number of USPs and bound ATPs, i.e., single USP, one USP bound to one ATP, two USPs alone, two USPs connected to one ATP and two ATPs.

**Figure 4 ijms-24-04725-f004:**
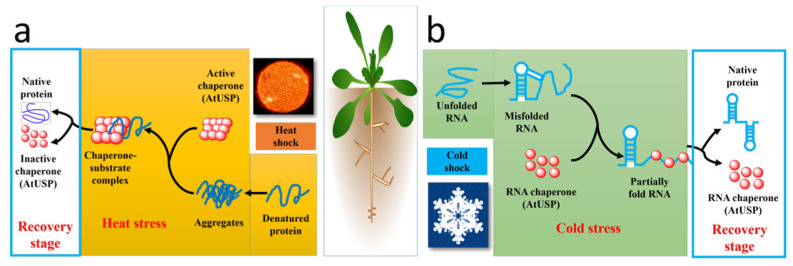
Role of Arabidopsis USPs in different stresses. (**a**) When the environment gives Arabidopsis high temperature stress, *At*USP functions as a protein chaperone in the cell, it changes from a low molecular weight complex (LMW) to a high molecular compound (HMW) and binds to proteins that cannot function properly because of high temperature aggregation, helping it to restore its active state after *At*USP reverts back to the LMW state. (**b**) When subjected to cold stress, *At*USP binds as an RNA chaperone to RNA fragments misfolded by low temperature and helps them to open the misfold and restore the correct fold.

**Table 1 ijms-24-04725-t001:** List of AtUSP genes in Arabidopsis thaliana along with their corresponding proteins, locus ID, CDS length, protein length, domain and Pfam ID. (Adapted with permission from Ref. [35]. 2023, copyright Springer Nature. and NCBI: https://www.ncbi.nlm.nih.gov/genome/ (accessed on 29 January 2023).

Gene	Protein	Locus	CDS Length (bp)	Protein Length (aa)	Domain	Pfam ID
*AtUSP1*	AtUSP1	At1G01680	927	308	USP	PF00582
*AtUSP2*	AtUSP2	At1G09740	516	171	USP	PF00582
*AtUSP3*	AtUSP3	At1G11360	729	242	USP	PF00582
*AtUtyK1*	AtUtyK1	At1G16760	2277	758	USP	PF00582
					Pkinase-Tyr	PF07714
*AtUtyK2*	AtUtyK2	At1G17540	2187	728	USP	PF00582
					Pkinase-Tyr	PF00069
*AtUK1*	AtUK1	At1G21590	2271	756	USP	PF00582
					Pkinase	PF00069
*AtUSP4*	AtUSP4	At1G44760	642	213	USP	PF00582
*AtUSP5*	AtUSP5	At1G48960	660	219	USP	PF00582
*AtUSP6*	AtUSP6	At1G68300	483	160	USP	PF00582
*AtUSP7*	AtUSP7.1	At1G69080.1	672	223	USP	PF00582
	AtUSP7.2	At1G69080.2	630	209	USP	PF00582
*AtUK2*	AtUK2	At1G77280	2385	753	USP	PF00582
					Pkinase	PF00069
*AtUtyK3*	AtUtyK3	At1G78940.2	2265	754	USP	PF00582
					Pkinase-Tyr	PF07714
*AtUSP8*	AtUSP8	At2G03720	498	165	USP	PF00582
*AtUK3*	AtUK3	At2G07020	2103	700	USP	PF00582
					Pkinase	PF00069
*AtUSP9*	AtUSP9.1	At2G21620.1	564	187	USP	PF00582
	AtUSP9.2	At2G21620.2	582	193	USP	PF00582
*AtUK4*	AtUK4	At2G24370	2367	788	USP	PF00582
					Pkinase	PF00069
*AtUSP10*	AtUSP10	At2G47710	489	162	USP	PF00582
*AtUSP11*	AtUSP11	At3G01520	528	175	USP	PF00582
*AtUSP12*	AtUSP12	At3G03270.1	606	201	USP	PF00582
*AtUSP13*	AtUSP13	At3G03290	825	274	USP	PF00582
*AtUSPUSP1*	AtUSPUSP1.1	At3G11930.1	600	199	USP	PF00582
	AtUSPUSP1.2	At3G11930.2	603	200	USP	PF00582
	AtUSPUSP1.3	At3G11930.3	681	226	USP, USP	PF00582
	AtUSPUSP1.4	At3G11930.4	606	201	USP	PF00582
*AtUSP14*	AtUSP14	At3G17020	492	163	USP	PF00582
*AtUtyK4*	AtUtyK4	At3G20200	2343	780	USP	PF00582
					Pkinase	PF00069
*AtUSP15*	AtUSP15	At3G21210	2415	804	USP	PF00582
*AtUSP16*	AtUSP16	At3G25930	465	154	USP	PF00582
*AtUSP17*	AtUSP17.1	At3G53990.1	483	160	USP	PF00582
	AtUSP17.2	At3G53990.2	381	126	USP	PF00582
*AtUSP18*	AtUSP18.1	At3G58450.1	615	204	USP	PF00582
	AtUSP18.2	At3G58450.2	594	197	USP	PF00582
*AtUSP19*	AtUSP19	At3G62550	489	162	USP	PF00582
*AtUtyK5*	AtUtyK5	At4G25160	2508	835	USP	PF00582
					Pkinase-Tyr	PF07714
*AtUSP20*	AtUSP20	AT4G13450.1	660	219	USP	PF00582
*AtUSP21*	AtUSP21	At4G27320	783	260	USP	PF00582
*AtUK5*	AtUK5	At4G31230	2295	764	USP	PF00582
					Pkinase	PF00069
*AtUtyK6*	AtUtyK6	At5G12000	2106	701	USP	PF00582
					Pkinase-Tyr	PF07714
*AtUSP22*	AtUSP22	At5G14680	528	175	USP	PF00582
*AtUSP23*	AtUSP23	At5G17390	858	285	USP	PF00582
*AtUSP24*	AtUSP24	At5G20310	1185	394	USP	PF00582
*AtUK6*	AtUK6	At5G35380	2196	731	USP	PF00582
					Pkinase	PF00069
*AtUSP25*	AtUSP25.1	At5G47740.1	735	244	USP	PF00582
	AtUSP25.2	At5G47740.2	741	246	USP	PF00582
*AtUSP26*	AtUSP26	At5G49050	453	150	USP	PF00582
*AtUSP27*	AtUSP27	At5G54430	729	242	USP	PF00582
*AtUK7*	AtUK7	At5G63940	2118	705	USP	PF00582
					Pkinase	PF00069

**Table 2 ijms-24-04725-t002:** Summary of crystal structures of USP family proteins.

Organism	PDB Code	Chains	Mutation(s)	Small Molecules	References
*Methanocaldococcus jannaschii*	1MJH	2	0	ATP, Mn^2+^	[57]
*Haemophilus influenzae*	1JMV	4	0	SO_4_^2−^	[58]
*Methanocaldococcus jannaschii*	5AHW	6	0	CMP, SO_4_^2−^, POG, Cl^−^	[59]
*Salmonella enterica subsp. enterica serovar Typhimurium str.* LT2	4R2J	1	0	GLC, PO_4_^3−^, EDO, Zn^2+^	[60]
	4R2K	1	1 (4R2J Mutation)	SO_4_^2−^, EDO, OXD	
	4R2L	2	0	EDO, Cl^−^, ATP, Mg^2+^	
	4R2M	2	1 (4R2L Mutation)	ANP, Mg^2+^	
*Nitrosomonas europaea* ATCC 19718	3TNJ	1	0	AMP	[61]
	2PFS	1	3	MSE, Cl^−^	
*Archaeoglobus fulgidus*	3QTB	2	0	D5M, ACT, MSE	
	6HCD	4	0	ACT, MSE, Cl^−^	
*Escherichia coli* K-12	5CB0	2	0	Z6X	[63]
*Arabidopsis thaliana*	2GM3	6	3	AMP, MSE	[64]

**Table 3 ijms-24-04725-t003:** Summary of the functions of USPs in different plants.

Plant Species	Name of USPs	Functions	References
*Arabidopsis thaliana*	AtUSP	Protein chaperone	[85]
		RNA chaperone	[90]
		Inherently antifungal activity	[55]
	AT5G35380	Related to resistance to flooding	[86]
	GRUSP	Promote seed germination	[36]
		Novel regulatory components of the flowering signal transduction pathway	[40]
	HRU1	Coordination of oxygen sensing and ROS signaling under hypoxic conditions	[87]
	AtUSP17	Related to the salt tolerance of the plant	[42]
*Gossypium arboretum*	GUSP1	Activates downstream genes in response to drought	[49,50]
	GUSP2	Directly involved in stress tolerance or as signaling molecules to activate stress adaptation mechanisms	[51]
*Malus sieversii*	MsUSPA	Reducing transpiration and retaining water by altering the cellular structure of the leaves to improve drought resistance	[46]
		Increase the activity of antioxidant enzymes, reduce the accumulation of Ros and improve the antioxidant capacity	
*Solanum lycoperiscus*	SlRd2	Regulation of SlCipk6-mediated ROS generation	[102]
*Solanum pennellii*	SpUSP	Improving tolerance to oxidative stress	[44]
*Salicornia brachiata*	SbUSP	Participate in tolerance mechanisms or act as molecular switches (signaling molecules) to activate stress adaptation mechanisms	[53]
*Sulfolobus acidocaldarius*	SaUspA	Binds phosphatase and alters phosphatase activity	[103]

## Data Availability

Not Applicable.

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
