# Peer review of "Universal Stress Proteins: From Gene to Function"

_ijms, 2023, doi:10.3390/ijms24054725_

Round 1
Reviewer 1 Report
The manuscript entitled "Universal Stress Proteins: from Gene to Function" is a well written and informative review paper in the field of universal stress proteins presenting the current knowledge about USPs and the future perspectives. I propose it should be accepted after minor revisions. My points are:
1. All organism names should be written in Italics.
2. Authors should check the gaps between words (some double, some without)
3. Line 28, there is a missing n
4. Line 56 H2O2 must be corrected
5. LIne 71 the first letter is missing
6. Lines 79-81 are a duplicate of lines 77-79 with another number of references. Please correct it and check the number of references and the right order.
7. Fig 4 needs a reference and as I consider a major proof of regulatory activity of certain proteins, it should be discussed with more details in the main text.
8. Line 300, please correct CO2
Author Response
Journal: International Journal of Molecular Sciences
Manuscript ID: IJMS-2217599
Title: “Universal stress proteins: from gene to function”
Author(s): Dan Luo, Zilin Wu, Qian Bai,Yong Zhang, Min Huang, Yajiao Huang, Xiangyang Li*
Dear Reviewers.
On behalf of my co-authors, I would like to thank you very much for giving us an opportunity to revise our manuscript, and we appreciate your positive and constructive comments and suggestions on our manuscript.
We have carefully revised the manuscript based on your comments. Each comment has been taken seriously and answered individually with reference to the relevant section of the manuscript. All corrections were made on the basis of the previous document and are highlighted in "red color" in the revised version. We hope that the corrections will be accepted. We have responded to your comments point by point and itemized the changes as follows.
Your Comments:
- The manuscript entitled "Universal Stress Proteins: from Gene to Function" is a well written and informative review paper in the field of universal stress proteins presenting the current knowledge about USPs and the future perspectives. I propose it should be accepted after minor revisions.
Ans.: Thank you for your affirmation of this manuscript, and we will fully incorporate your comments.
- All organism names should be written in Italics.
Ans.: Thank you for your kind suggestions. Based on your suggestion we have italicized the names of the organisms involved in the manuscript and marked them in red in the manuscript.
- Authors should check the gaps between words (some double, some without).
Ans.: Thanks very much for your kind suggestions., We have done a detailed check of the manual operation and corrected the problems you mentioned.
- Line 28, there is a missing n.
Ans.: Thanks very much for your kind suggestions. We have revised the issue and shown it in red in the manuscript.
- Line 56 H2O2 must be corrected.
Ans.: Thanks very much for your kind suggestions. We have revised the issue and shown it in red in the manuscript.
- Line 71 the first letter is missing.
Ans.: Thanks very much for your kind suggestions. We have revised the issue and shown it in red in the manuscript.
- Lines 79-81 are a duplicate of lines 77-79 with another number of references. Please correct it and check the number of references and the right order.
Ans.: Thanks very much for your kind suggestions. We have made the changes you suggested, which are located on lines 84-86 of the manuscript and are shown in red.
- Fig 4 needs a reference and as I consider a major proof of regulatory activity of certain proteins, it should be discussed with more details in the main text.
Ans.: Thanks very much for your kind suggestions. We have made the changes you suggested, which are located on lines 259-264 and 275-279 of the manuscript and are shown in red.
“AT3G53990 has several possible functions in Arabidopsis. AT3G53990 transforms itself from a low molecular weight (LMW) complex to a high molecular weight (HMW) complex in response to high temperature scorching stress, and such a structural change could help plants protect key intracellular proteins in high temperature environments, and it would participate as a protein chaperone to play a stabilizing role as a protein.”
“As important stress-regulated proteins, USPs are involved in a range of physiological activities in plants. It has been shown that cold-shock tolerance is usually achieved by RNA chaperones [88], and that, in Arabidopsis, AtUSP becomes involved in plant physiological activities as an RNA chaperone, which helps RNA-bound proteins to exercise their functions correctly by preventing RNA misfolding or by resolving misfolded RNAs”.
- Line 300, please correct CO2.
Ans.: Thanks very much for your kind suggestions. We have revised the issue and shown it in red in the manuscript.
Thank you and best regards.
Prof. & Dr. Xiangyang Li
State Key Laboratory Breeding Base of Green Pesticide and Agricultural Bioengineering, Key Laboratory of Green Pesticide and Agricultural Bioengineering, Ministry of Education, Guizhou University, Huaxi District, Guiyang 550025, PR China.
E-mail: xyli1@gzu.edu.cn (X.Y.L.).

Reviewer 2 Report
The proposed review titled “Universal Stress Proteins: from Gene to Function” has been systematically written and can be accepted for publication after minor revision. I believe such informative review will help researchers to do further research.
[1] The abstract of the manuscript still requires some details. The exact role of USPs in plants and their significance is still missing.
[2] Authors must explain in the introduction what is missing in already published articles and what they are offering in this review article.
[3] Section 3. The relationship between USP crystal structure and properties can be discussed here.
[4] Sections 4.1 and 4.2 are still confusing as they carry the same title.
[5] Section 5. More information regarding future research direction can be provided here.
[6] Some typing errors can be avoided.
Author Response
Journal: International Journal of Molecular Sciences
Manuscript ID: IJMS-2217599
Title: “Universal stress proteins: from gene to function”
Author(s): Dan Luo, Zilin Wu, Qian Bai, Yong Zhang, Min Huang, Yajiao Huang, Xiangyang Li*
Dear Reviewers.
On behalf of my co-authors, I would like to thank you very much for giving us an opportunity to revise our manuscript, and we appreciate your positive and constructive comments and suggestions on our manuscript.
We have carefully revised the manuscript based on your comments. Each comment has been taken seriously and answered individually with reference to the relevant section of the manuscript. All corrections were made on the basis of the previous document and are highlighted in "red color" in the revised version. We hope that the corrections will be accepted. We have responded to your comments point by point and itemized the changes as follows.
Your Comments:
- The proposed review titled “Universal Stress Proteins: from Gene to Function” has been systematically written and can be accepted for publication after minor revision. I believe such informative review will help researchers to do further research.
Ans.: Thank you for your affirmation of this manuscript, and we will fully incorporate your comments.
- The abstract of the manuscript still requires some details. The exact role of USPs in plants and their significance is still missing.
Ans.: Thank you for your kind suggestions. We have revised your suggestion in detail and added a table showing the different roles played by USPs in different plants, which is located in line 358 of the manuscript and is shown in red.
Table 3. Summary of the functions of USPs in different plants
|
Plant species |
Name of USPs |
Functions |
References |
|
Arabidopsis thaliana |
AtUSP |
Protein chaperone |
[85] |
|
RNA chaperone |
[90] |
||
|
Inherently antifungal activity |
[55] |
||
|
AT5G35380 |
Related to resistance to flooding |
[86] |
|
|
GRUSP |
Promote seed germination |
[36] |
|
|
Novel regulatory components of the flowering signal transduction pathway |
[40] |
||
|
HRU1 |
Coordination of oxygen sensing and ROS signaling under hypoxic conditions |
[87] |
|
|
AtUSP17 |
Related to the salt tolerance of the plant |
[42] |
|
|
Gossypium arboretum |
GUSP1 |
Activates downstream genes in response to drought |
[49, 50] |
|
GUSP2 |
Directly involved in stress tolerance or as signaling molecules to activate stress adaptation mechanisms |
[51] |
|
|
Malus sieversii |
MsUSPA |
Reducing transpiration and retaining water by altering the cellular structure of the leaves to improve drought resistance |
[46] |
|
Increase the activity of antioxidant enzymes, reduce the accumulation of Ros and improve the antioxidant capacity |
|||
|
Solanum lycoperiscus |
SlRd2 |
Regulation of SlCipk6-mediated ROS generation |
[102] |
|
Solanum pennellii |
SpUSP |
Improving tolerance to oxidative stress |
[44] |
|
Salicornia brachiata |
SbUSP |
Participate in tolerance mechanisms or act as molecular switches (signaling molecules) to activate stress adaptation mechanisms |
[53] |
|
Sulfolobus acidocaldarius |
SaUspA |
Binds phosphatase and alters phosphatase activity |
[103] |
- Authors must explain in the introduction what is missing in already published articles and what they are offering in this review article.
Ans.: Thanks very much for your kind suggestions. We have made the changes you suggested, which are located on lines 53-58 of the manuscript and are shown in red.
The study of USPs in plants is not as in-depth as in microorganisms, and most of the studies on plant USPs focus on plant resistance studies, with few studies on structure and other aspects. In this paper, we summarize the studies on USPs from three aspects: genetic, structural and functional, in order to provide a reference for the study of USPs in plants.
- Section 3. The relationship between USP crystal structure and properties can be discussed here.
Ans.: Thanks very much for your kind suggestions. We have made the changes you suggested, which are located on lines 187-192 of the manuscript and are shown in red.
“It is well known that structure determines function, and by combing through them we found that most USPs structures rarely exist as single chains when they are resolved, (Table 2) they are often aggregated in even chains, while ATP is involved in protein crystallization, which may be an important reason for its function in plants.”
Table 2. Summary of crystal structures of USP family proteins
|
Organism |
PDB code |
Chains |
Mutation(s) |
Small molecules |
References |
|
Methanocaldococcus jannaschii |
1MJH |
2 |
0 |
ATP、Mn2+ |
[57] |
|
Haemophilus influenzae |
1JMV |
4 |
0 |
SO42- |
[58] |
|
Methanocaldococcus jannaschii |
5AHW |
6 |
0 |
CMP、 SO42-、POG、 Cl- |
[59] |
|
Salmonella enterica subsp. enterica serovar Typhimurium str. LT2 |
4R2J |
1 |
0 |
GLC、 PO43-、EDO、Zn2+ |
[60] |
|
4R2K |
1 |
1(4R2J Mutation) |
SO42-、 EDO、 OXD |
||
|
4R2L |
2 |
0 |
EDO、 Cl-、ATP、Mg2+ |
||
|
4R2M |
2 |
1(4R2L Mutation) |
ANP、Mg2+ |
||
|
Nitrosomonas europaea ATCC 19718 |
3TNJ |
1 |
0 |
AMP |
[61] |
|
2PFS |
1 |
3 |
MSE、Cl- |
||
|
Archaeoglobus fulgidus |
3QTB |
2 |
0 |
D5M、ACT、MSE |
|
|
6HCD |
4 |
0 |
ACT、MSE、Cl- |
||
|
Escherichia coli K-12 |
5CB0 |
2 |
0 |
Z6X |
[63] |
|
Arabidopsis thaliana |
2GM3 |
6 |
3 |
AMP、MSE |
[64] |
- Sections 4.1 and 4.2 are still confusing as they carry the same title.
Ans.: Thanks very much for your kind suggestions. Section 4.1, entitled “4.1. Functional diversity of prokaryotic USPs”, describes the role of USPs in prokaryotic organisms, while section 4.2, entitled “4.2. Functional diversity of eukaryotic USPs”, describes the role of USPs in eukaryotic organisms.
- Section 5. More information regarding future research direction can be provided here.
Ans.: Thanks very much for your kind suggestions. We have made the changes you suggested, which are located on lines 354-357 of the manuscript and are shown in red.
“USPs from different plants, which play different roles in vivo (Table 3), are also worthy of research in the future by genetically engineering plants to adapt them to more complex climatic challenges.”
- Some typing errors can be avoided.
Ans.: Thanks very much for your kind suggestions. We have revised the issue and shown it in red in the manuscript.
Thank you and best regards.
Prof. & Dr. Xiangyang Li
State Key Laboratory Breeding Base of Green Pesticide and Agricultural Bioengineering, Key Laboratory of Green Pesticide and Agricultural Bioengineering, Ministry of Education, Guizhou University, Huaxi District, Guiyang 550025, PR China.
E-mail: xyli1@gzu.edu.cn (X.Y.L.).

Reviewer 3 Report
The article by Luo et at deals with an interesting topic. USPs proteins, vital for survival under stressful conditions of many species, are undoubtedly a topic worthy of a review. However, the article, in its present form, requires a proofreading effort from the authors. There are many typos. Some examples:
Lane 34 cites panel C before panel B. I suggest to change the order of the panels in the figure
Lane 28: “phosphorylation” should be phosphorylation
Lane 38: figure caption 1 uspaC should be uspC according to the figure
Lane 56: “H2O2”. The numbers should be in the subscript form.
Lane 71: he model plant should be The model
Lane 72: Barley, for example, has a tissue-specific HvUsp gene that is specifically expressed in the roots and leaves of barley repeats two times the word barley.
Lane 81: “in the plan” should be plant.
Lane 148: “.....with 222 symmetries”. there is no need to add the number of symmetries.
Lane 171: “....it consists of three At3g01520 protein dimers, each protomer having one AMP molecule bound to it [64]”. Consider rephrasing.
Lane 172: “The crystalline small molecules containing ATP or ATP analogues…”. This is unclear. Consider rephrasing. The same sentence should be fixed also in the figure caption.
Figure 2 caption: consider rephrasing the first sentence and lane 178 that repeats twice five structures-five structures.
Figure 3 caption: consider rephrasing the last sentence.
The statement on line 71 does not agree with the statement on line 244.
Moreover, the Latin names of all the organism mentioned in the review (i.e. A. thaliana or M. tuberculosis) should be in Italics.
Also, authors should be careful when naming genes or proteins: generally, the name in italics is used to indicate the gene, while only the first letter is capitalized and the rest lowercase for proteins (no italics).
Author Response
Journal: International Journal of Molecular Sciences
Manuscript ID: IJMS-2217599
Title: “Universal stress proteins: from gene to function”
Author(s): Dan Luo, Zilin Wu, Qian Bai,Yong Zhang, Min Huang, Yajiao Huang, Xiangyang Li*
Dear Reviewers.
On behalf of my co-authors, I would like to thank you very much for giving us an opportunity to revise our manuscript, and we appreciate your positive and constructive comments and suggestions on our manuscript.
We have carefully revised the manuscript based on your comments. Each comment has been taken seriously and answered individually with reference to the relevant section of the manuscript. All corrections were made on the basis of the previous document and are highlighted in "red color" in the revised version. We hope that the corrections will be accepted. We have responded to your comments point by point and itemized the changes as follows.
Your Comments:
- The article by Luo et at deals with an interesting topic. USPs proteins, vital for survival under stressful conditions of many species, are undoubtedly a topic worthy of a review.
Ans.: Thank you for your affirmation of this manuscript, and we will fully incorporate your comments.
- Lane 34 cites panel C before panel B. I suggest to change the order of the panels in the figure
Ans.: Thank you for your kind suggestions. We have made the changes you suggested, which are located on lines 32-35 of the manuscript and are shown in red. “and they have been divided into three subfamilies (Figure 1a) [11]. Various E. coli USPs play different roles in this bacterium [12] and can be divided into two major classes (Figure 1b) and four minor subclasses (Figure 1c) [13].”
- Lane 28: “phosphorylation” should be phosphorylation
Ans.: Thanks very much for your kind suggestions. We have revised the issue and shown it in red in the manuscript.
- Lane 38: figure caption 1 uspaC should be uspC according to the figure
Ans.: Thanks very much for your kind suggestions. We have revised the issue and shown it in red in the manuscript.
- Lane 56: “H2O2”. The numbers should be in the subscript form.
Ans.: Thanks very much for your kind suggestions. We have revised the issue and shown it in red in the manuscript.
- Lane 71: he model plant should be The model
Ans.: Thanks very much for your kind suggestions., We have done a detailed check of the manual operation and corrected the problems you mentioned.
- Lane 72: Barley, for example, has a tissue-specific HvUsp gene that is specifically expressed in the roots and leaves of barley repeats two times the word barley.
Ans.: Thanks very much for your kind suggestions. We have made the changes you suggested, which are located on lines 79-80 of the manuscript and are shown in red.
“For example, barley has HvUsp genes that are specifically expressed only in roots and leaves”.
- Lane 81: “in the plan” should be plant.
Ans.: Thanks very much for your kind suggestions. We have revised the issue and shown it in red in the manuscript.
- Lane 148: “.....with 222 symmetries”. there is no need to add the number of symmetries.
Ans.: Thanks very much for your kind suggestions. We have made the changes you suggested, which are located on lines 150-152 of the manuscript and are shown in red.
“The two YnaF plasmids form an asymmetric unit that is stabilized by interaction with chloride ions to form a tetramer with symmetry”.
- Lane 171: “....it consists of three At3g01520 protein dimers, each protomer having one AMP molecule bound to it [64]”. Consider rephrasing.
Ans.: Thanks very much for your kind suggestions. We have made the changes you suggested, which are located on lines172-175 of the manuscript and are shown in red.
“The Arabidopsis protein At3g01520 (2GM3, PDB DOI: 10.2210/pdb2GM3/pdb) is the only eukaryotic universal stress protein crystal structure that has been resolved, and the structure shows that it is an aggregate and that each monomer is bound to an AMP molecule its [64].”
- Lane 172: “The crystalline small molecules containing ATP or ATP analogues…”. This is unclear. Consider rephrasing. The same sentence should be fixed also in the figure caption.
Ans.: Thanks very much for your kind suggestions. We have made the changes you suggested, which are located on lines176 and 181 of the manuscript and are shown in red.
“ATP or ATP analogs (AMP, ANP, etc.)”
- Figure 2 caption: consider rephrasing the first sentence and lane 178 that repeats twice five structures-five structures.
Ans.: Thanks very much for your kind suggestions. We have made the changes you suggested, which are located on lines180-186 of the manuscript and are shown in red.
“Figure 2. USPs binding analysis. Crystalline small molecules containing ATP or ATP analogs (AMP, ANP, etc.) were selected for sequence comparison and struc-ture superimposition. (a) Sequence comparison. all except 4R2L contain ATP-binding motifs: G-2X-G-9X-S/T. (b) Superposition. 1MJH, 5AHW, 4R2L, 3TNJ and 2GM3 are gold, blue, purple, green and red, respectively. Superposition of 1MJH as the reference structure revealed that 4R2L still binds ATP at the same position even without the typical ATP binding pattern.”
- Figure 3 caption: consider rephrasing the last sentence.
Ans.: Thanks very much for your kind suggestions. We have made the changes you suggested, which are located on lines224-227 of the manuscript and are shown in red.
“(c) Classification of the USP of Vibrio desulfuricans. The USPs can be classified into five categories according to the number of USPs and bound ATPs, i.e., single USP, one USP bound to one ATP, two USPs alone, two USPs connected to one ATP and two ATPs.”
- The statement on line 71 does not agree with the statement on line 244.
Ans.: Thanks very much for your kind suggestions. We have done a detailed check of the manuscript and corrected the issues you mentioned.
- Moreover, the Latin names of all the organism mentioned in the review (i.e. thaliana or M. tuberculosis) should be in Italics.
Ans.: Thank you for your kind suggestions. Based on your suggestion we have italicized the names of the organisms involved in the manuscript and marked them in red in the manuscript.
- Also, authors should be careful when naming genes or proteins: generally, the name in italics is used to indicate the gene, while only the first letter is capitalized and the rest lowercase for proteins (no italics).
Ans.: Thank you for your kind suggestions. Based on your suggestion we have italicized the names of the organisms involved in the manuscript and marked them in red in the manuscript.
Thank you and best regards.
Prof. & Dr. Xiangyang Li
State Key Laboratory Breeding Base of Green Pesticide and Agricultural Bioengineering, Key Laboratory of Green Pesticide and Agricultural Bioengineering, Ministry of Education, Guizhou University, Huaxi District, Guiyang 550025, PR China.
E-mail: xyli1@gzu.edu.cn (X.Y.L.).
